# Avoidable Blood Loss in Critical Care and Patient Blood Management: Scoping Review of Diagnostic Blood Loss

**DOI:** 10.3390/jcm11020320

**Published:** 2022-01-10

**Authors:** Philipp Helmer, Sebastian Hottenrott, Andreas Steinisch, Daniel Röder, Jörg Schubert, Udo Steigerwald, Suma Choorapoikayil, Patrick Meybohm

**Affiliations:** 1Department of Anaesthesiology, Intensive Care, Emergency and Pain Medicine, University Hospital Würzburg, Oberdürrbacher Str. 6, 97080 Wuerzburg, Germany; helmer_p@ukw.de (P.H.); hottenrott_s@ukw.de (S.H.); Steinisch_a@ukw.de (A.S.); roeder_d@ukw.de (D.R.); 2Department of Laboratory Medicine and Coagulation Ambulance, University Hospital Würzburg, Oberdürrbacher Str. 6, 97080 Würzburg, Germany; schubert_j@ukw.de (J.S.); steigerwald_u@ukw.de (U.S.); 3Department of Anaesthesiology, Intensive Care Medicine and Pain Therapy, University Hospital Frankfurt, Goethe-University, 60323 Frankfurt, Germany; Suma.Choorapoikayil@kgu.de

**Keywords:** PBM, critically ill, intensive care, iatrogenic anemia, avoidable blood loss, diagnostic blood loss

## Abstract

Background: Anemia remains one of the most common comorbidities in intensive care patients worldwide. The cause of anemia is often multifactorial and triggered by underlying disease, comorbidities, and iatrogenic factors, such as diagnostic phlebotomies. As anemia is associated with a worse outcome, especially in intensive care patients, unnecessary iatrogenic blood loss must be avoided. Therefore, this scoping review addresses the amount of blood loss during routine phlebotomies in adult (>17 years) intensive care patients and whether there are factors that need to be improved in terms of patient blood management (PBM). Methods: A systematic search of the Medline Database via PubMed was conducted according to PRISMA guidelines. The reported daily blood volume for diagnostics and other relevant information from eligible studies were charted. Results: A total of 2167 studies were identified in our search, of which 38 studies met the inclusion criteria (9 interventional studies and 29 observational studies). The majority of the studies were conducted in the US (37%) and Canada (13%). An increasing interest to reduce iatrogenic blood loss has been observed since 2015. Phlebotomized blood volume per patient per day was up to 377 mL. All interventional trials showed that the use of pediatric-sized blood collection tubes can significantly reduce the daily amount of blood drawn. Conclusion: Iatrogenic blood loss for diagnostic purposes contributes significantly to the development and exacerbation of hospital-acquired anemia. Therefore, a comprehensive PBM in intensive care is urgently needed to reduce avoidable blood loss, including blood-sparing techniques, regular advanced training, and small-volume blood collection tubes.

## 1. Introduction

### 1.1. Anemia in Critically Ill Patients

Anemia is a common comorbidity in intensive care patients with a prevalence of up to 98% [1], of which 40–50% suffer from severe anemia (hemoglobin < 9 g/dL) [2]. Several studies have repeatedly demonstrated that anemia increases the administration of allogeneic blood products and is furthermore associated with an increased complication rate, prolonged hospital stay, and increased mortality rate [3,4]. Particularly, in patients with COVID-19 infection, mortality rate increases significantly in the presence of anemia, from 8% to 25%, compared to COVID-19 patients without anemia [5]. McEvoy et al. revealed that anemia increases the 90 day mortality rate in patients with chronic obstructive pulmonary disease (COPD) from 25% to 57% [6,7]. The authors also revealed that, in patients with congestive heart failure, acute myocardial infarction, or chronic kidney disease, an increased rate of adverse events was observed in anemic patients compared to non-anemic patients [6]. It is noteworthy to mention that critically ill anemic patients often show a prolonged hospital stay after discharge from ICU [8]. Cioc et al. evaluated the impact of daily phlebotomy in intensive care patients on hemoglobin levels and found that both the absolute hemoglobin value and dynamics of the hemoglobin drop play an important role. The difference between hemoglobin values at admission to an ICU and at discharge is associated with an increased overall mortality [9]. The mainstay of anemia treatment in critically ill patients is transfusion of allogenic red blood cell (RBC) units. Up to 85% of the patients with an ICU stay >7 days received at least one unit of RBC [10]. 

### 1.2. Pathophysiology of Anemia in Critically Ill Patients

The causes of anemia are multifactorial, including pathophysiological and iatrogenic factors. Healthy adults produce 0.25 mL/kg RBCs per day, resulting in 0.5 L blood per week, whereas in critically ill patients, erythropoiesis is often impaired, leading to a reduced amount of newly produced RBCs [6]. Table 1 summarizes iatrogenic and pathophysiological factors that are associated with the development of anemia in critically ill patients [2,11,12].

Erythropoiesis is impaired in critically ill patients and results in reduced RBC production [13,14] due to the reduced life span of RBCs, elevated expression of inflammatory cytokines (TNF-alpha and Interleukin 1) [2,15,16], endogenous kidney dysfunction (leading to decreased serum erythropoietin (EPO) concentrations) [1], and reduced response from reticulocytes to EPO [1]. The administration of erythropoiesis-stimulating agents (ESA) can compensate for absolute or relative EPO deficiency, improving Hb levels in intensive care patients [13,17]. Almost 40% of ICU patients show changes in iron metabolism, 9% nutritional iron deficiency, and 2% low folate or Vitamin B12 [6,18,19,20]. In addition, hemolysis, particularly in patients with ARDS, sepsis, and post-administration of RBC units [21], fluid resuscitation (leading to dilutional anemia) [2,22], major hemorrhages (in 8.4%), and occult hemorrhages (in 90%) [1,6,23,24,25] can aggravate anemia by coagulation disorders [26] or by drug-induced changes in blood coagulation, e.g., by administration of anticoagulants. 

Iatrogenic factors include all types of medical interventions associated with potential blood loss or factors contributing to impaired erythropoiesis. The impact of iatrogenic anemia was highlighted in the early 1970s as nosocomial anemia but is still a persisting problem [27,28,29]. Corwin et al. demonstrated an association between the daily volume of diagnostic blood draws, length of stay, and the number of allogenic RBC units [10]. In case the total volume for diagnostic blood draws increased to 2156 ± 208 mL, patients required more than 10 units of RBCs compared to 601 ± 77 mL blood loss with no transfused RBC units. In cardiac surgical patients, diagnostic blood sampling resulted in 653 mL blood loss, which corresponds to 1–2 RBC units [28]. In individual cases of very long hospital stays of more than 100 days, the volume of blood collected even exceeded 12 L [28]. Interestingly, even small increases of daily phlebotomies doubled the likelihood of allogenic blood transfusions after day 21 [30]. 

In addition to the blood used for diagnostic purposes, high volumes of blood are often discarded during blood withdrawal if blood-saving collection systems are not used. Importantly, clinical trials in patients requiring intensive care can also contribute to iatrogenic anemia. Depending on the study design, PETAL network recommended a maximal research blood volume of 120 mL/d [31]. The estimated blood loss is much higher in ICU patients than in a surgical ward (up to 10 mL/day) or in renal inpatients [32,33]. Particularly on the day of ICU admission, the volume of daily blood collected for diagnostic purposes is the highest and decreases during the ICU stay [1,33]. 

### 1.3. Patient Blood Management

Based on the possibilities to strengthen and preserve patients’ own blood mass and to enable safe handling of donor blood, the World Health Assembly (WHA 63.12) [34] endorsed patient blood management (PBM) and called on the World Health Organization in 2010 to provide training to its member states on the safe and rational use of allogeneic blood products and implementation of transfusion alternatives. 

PBM is an interdisciplinary concept based on three pillars, with the aim to increase patient safety using diagnostic, therapeutic, and behavioral concepts. More than 100 specific measures [35] were designed to (1) provide comprehensive anemia management, (2) minimize iatrogenic (unnecessary) blood loss, and (3) harness and optimize the patient-specific physiological tolerance of anemia [36]. To reduce iatrogenic blood loss, PBM includes the use of autologous blood recovery systems, restrictive frequency of blood draws, use of reduced blood volume collection tubes and blood-sparing techniques, and accurate documentation of all blood losses. This scoping review focuses on iatrogenic anemia caused by phlebotomy and possible measures to reduce blood loss during diagnostics in critical care patients. 

## 2. Materials and Methods

### 2.1. Study Design 

This scoping review was reported according to the recommendations of PRISMA using Medline for systematic search [37]. No study protocol was published. Eligibility criteria were defined prior to literature search. All randomized or nonrandomized controlled trials, observational, prospective, or retrospective studies reporting daily volume of blood taken for laboratory diagnostics in an adult ICU were included. Expert opinions and individual case reports were excluded. Furthermore, reference lists of scooping reviews, systematic reviews, and meta analyses were screened. Trials including patients who were <18 years, non-human species, non English or German language, or for whom no full text was available even after contacting the authors were excluded from analysis. There were no specifications for length of stay on an ICU. 

### 2.2. Information Sources

The literature search was conducted in August 2021. The search term was compiled jointly by the authors after extensive discussions before the systematic literature search started. The search was deliberately based exclusively on keywords in the title and abstract in order to specify the search. Titles with unclear relevance were discussed in detail by the reviewers. All relevant information of the identified studies was discussed and independently summarized by two reviewers. 

### 2.3. Search Term

The search term contained relevant terms on diagnostic blood loss, blood draws, and iatrogenic anemia in intensive care patients. This can be found in Appendix A. 

### 2.4. Data Items

The primary endpoint was the volume of daily blood draws and, if indicated, the cumulative volume of blood draws. Secondary endpoints were measures to avoid or reduce blood loss and the use of RBC units. For both the identified observational and intervention studies, in addition to the first author, the year of publication, the country, or region in which the study was conducted, the patient population under observation, including size of the cohort studied, the study design, and the length of time patients received intensive care therapy, was reported. 

## 3. Results

Based on the systematic search, 2149 titles were found in Medline with exclusive consideration of adult cohorts (>18 years) and a further 18 studies were found by a hand search, resulting in 2167 possible eligible studies. After screening title and abstracts, 2034 studies were excluded, and 133 studies were considered for full text search. A total of 94 studies were further excluded: 46 studies provided no explicit data on daily blood draws, 37 studies focused on other topics, 9 studies could not be translated for the purposes of this study, for 2 studies, no full text was available even after contacting the authors, and 1 study included non-human species. The remaining 38 studies were grouped according to the study design (observational and interventional) and analyzed separately (Figure 1). Details about included studies are shown in Table 2 and Table 3. 

A total of 29 observational studies and 9 interventional studies were included. Overall, 14 studies were published in the USA, 5 studies in Canada, 4 studies in the United Kingdom, 3 studies in Australia, and 2 studies in Germany. The age of the patients ranged from 46 to 72 years. Only three studies were randomized. In total, 18 studies had a prospective design, 15 studies a retrospective design, and 5 studies had no descriptions about study design. 

### 3.1. Interventional Trials

Overall, we found five interventional trials studying the effects of pediatric size blood collection tubes, three intervention trials comparing different blood conservation devices, and one study investigating a comprehensive blood saving bundle (Figure 2). All interventional trials showed that the use of pediatric sized blood collections tubes can significantly reduce the daily amount of blood drawn, for example in a surgical ICU from 240 mL/d to 150 mL/d [45]. In addition, one trial showed a significant effect of using pediatric tubes vs. adult sized tubes on mortality (5.5% vs. 1.5%) [42].

### 3.2. Observational Trials

The amount of daily mean diagnostic blood drawn varies between 9.8 ± 5.5 mL/d [51] and 377 mL/d [45]. Figure 3 presents the volume of daily diagnostic blood drawn depending on different types of ICUs. Koch et al. showed that patients on cardiothoracic ICUs lose a median of 332 mL during their stay [28]. Furthermore, Holland et al. showed high volumes of daily blood loss of 62 mL compared to non-cardiac surgery cohorts [49], suggesting that these cardiac patients are particularly at risk for the development of iatrogenic anemia. Vincent et al. found in >1000 patients in Western European ICUs an average blood loss for diagnostics of up to 41.1 mL/d [52]. Quinn et al. and Chornenki et al. revealed comparable amounts of blood loss in North American ICU patients and in Canada [33,57]. Most of the studies included data from either surgical or general ICUs. Data from one trial focusing on patients with acute myocardial infarction showed cumulative blood loss of up to 246 mL due to phlebotomy. The authors also revealed an independent correlation between the increased use of phlebotomy and the risk to develop hospital acquired anemia [29]. 

## 4. Discussion

Intensive care patients show a high prevalence of anemia associated with an increased demand for RBC transfusions. Therefore, the question arises as to what extent this problem can be addressed. Previous studies showed high variance in the amount of blood loss during the hospital stay. This scoping review confirmed these findings as blood loss due to diagnostic sampling ranged between 10 mL and 377 mL per day in ICU. In patients receiving intensive care for more than 7 days, hemoglobin decreased by −0.6 ± 0.6 g/dL during the first three days [11]. The median Hb decreased by about −10.2 ± 15% during the first week on ICU [33]. 

The most common performed laboratory tests are blood gas analysis (BGA) (40%), followed by a coagulation test (18%) and complete blood count (14%) [28]. Serum chemistries are associated with the highest amount of blood drawn per day [29]. The volume of blood typically needed for BGA is approximately 2 mL, for serum chemistries it is 7.5 mL, for citrate tubes it is 4.3 mL, for EDTA it is 2.7 mL, and it is up to 6 × 10–20 mL for blood cultures. A systematic review by Siegal et al. pointed out that only 10% of collected blood is used for laboratory tests, while the remaining blood drawn is discarded [66]. However, the authors encountered a problem in the quality of evidence of studies about blood loss for diagnostics and included only eight studies in their analysis, while >3000 studies were screened. Reasons for this are due to different methodologies of the studies, as no recognized standards for the design of comparable studies have been established yet. To a lesser extent, blood volumes are measured in the context of prospective studies, but most authors estimated or calculated their data. In retrospective trials, neither all blood samples nor discarded volumes were recorded. It should also be noted that, depending on the normal distribution, either only the median or only the mean value was given, and these values cannot be fully compared. Furthermore, there are also significant differences between different hospitals in terms of the definition of an ICU, medical care, and geographical location. Interestingly, teaching hospitals show more blood drawn compared to non-teaching hospitals [59]. Most studies do not distinguish between an ICU and the intermediate care unit, or there are no defined specifications about the severity of illness of the patients studied. In order to provide valid conclusions in the future and to be able to determine risk groups precisely, a standardized methodology of trials, including separate quantification of discarded volume and volume used for diagnostics, would be desirable. Furthermore, it would be beneficial to specify the hospital and ICU, as well as the patient cohort and to what extent PBM bundles have already been established. Patient characteristics should also be provided, including age, gender, and severity of illness using established scores (e.g., SOFA, APACHE II).

This scoping review shows that diagnostic blood collections can reach high volumes and is a major factor for iatrogenic induced anemia in ICU patients. Anemia is not limited to the immediate hospital stay. Even more than 6 months after discharge from an ICU, anemia is persistent in 53% of patients [67]. The consequences of anemia in this patient population are severe compared to non-intensive care patients [68]. In a retrospective study from Van der Laan et al., the presence of anemia at ICU discharge was an independent risk factor for a worse outcome [69]. Additionally, an observational study by Smilowitz et al., including >3000 patients, identified anemia as a predictor of reduced long-term survival [70]. Transfusion of RBC units is the mainstay to treat anemia [6]. However, on the one hand, patients with RBC transfusion requirements must be monitored more closely with regard to hemoglobin levels, leading to increased iatrogenic blood loss, while on the other hand, transfusion-dependent patients are usually sicker with impaired erythropoiesis compared to non-transfusion-dependent patients [28]. Therefore, some patients have an increased risk for iatrogenic anemia. Single multicenter studies identified the following risk groups suffering from anemia caused by blood loss: septic patients, patients with organ dysfunctions, transplanted patients, patients with renal failure (specifically renal replacement therapy), ventilated patients, patients receiving RBC transfusions, elderly persons, Jehovah’s Witnesses, and neonates [2,11,49,56,71,72]. Especially in septic patients, a hemoglobin decrease of −0.82 ± 0.81 g/dL per day within the first 3 days and a decrease of −0.3 g/dL per day for every further day was observed [11]. Long ICU treatment periods also result in higher volumes of diagnostic blood draws. Vincent et al. revealed a positive correlation between the severity of organ dysfunction defined as the SOFA-score and the frequency of blood sampling, resulting in increased iatrogenic blood loss in patients with worse medical conditions [52]. 

This leads to the question, which improvements can be implemented for blood-sparing techniques in intensive care? In mechanically ventilated patients, for example, blood-saving bundles could reduce diagnostic blood loss by 62% [44]. Considering that bedside waste per blood draw is 3.9 mL for arterial lines, 5.5 mL for central venous catheters, and 6.3 mL for peripheral venous catheters [56], the use of blood conservation devices is indicated. Siegal et al. showed that blood sampling conservation devices or a venous arterial blood management protection device (VAMP) can significantly reduce diagnostic blood loss by up to 80% [66]. The establishment of push-pull protocols is effective to significantly reduce the amount of discarded volume [73]. Dale et al. showed, almost 30 years ago, that the amount of blood for L-Thyroxin measurements in laboratory testing is about 0.06 mL but 5.9 mL was phlebotomized [58]. In addition to the discarded volume, the use of pediatric-size tubes can also reduce the amount of phlebotomized volume by up to 50% and provide reliable analytics [22,38,74,75]. In Figure 4, examples of standard size tubes and reduced sized tubes are displayed. As part of the reduction in tube filling volume, a total of approximately 40 L of patient blood could be saved per year (about 3700 treatment days) in a general surgical ICU with 12 beds at University Hospital Würzburg (Table 4). Calculated to the entire University Hospital Würzburg, a complete reduction in the filling volume of all blood tubes used would result in a saving of over 1600 L of patient blood per year.

In addition to the phlebotomized volume, an effort should also be made to decrease the frequency of blood sampling. Educational initiatives for reducing iatrogenic anemia should be considered, including all health care professionals and a feedback system to the ordering clinicians [76]. In addition, all laboratory values obtained should be shared with all medical disciplines to avoid redundant laboratory analyses. Furthermore, blood samples should be shared between the laboratories to reduce blood drawn. The indication for the determination of laboratory values should not be made according to strict standards but should be individually adapted to the respective patient [1]. This restricted indication should also apply to the placement of arterial line catheters. Low et al. has shown that patients with a comparable APACHE II score had 44% more phlebotomized blood volume/d if an arterial line was implemented [50]. Non-invasive measurement of glucose and hemoglobin levels by point-of-care diagnostics or continuous intravascular monitoring sensors should be considered to reduce the frequency of blood sampling [77,78]. 

## 5. Conclusions

Anemia may worsen the outcome of patients receiving intensive care. One of the most important iatrogenic factors aggravating anemia is blood sampling for diagnostic purposes. Vulnerable patients with acute myocardial infarction or pre-existing diseases, such as COPD, congestive heart failure, and chronic kidney disease, could benefit from patient centered PBM with reduced iatrogenic blood loss. Patient groups at risk for frequent blood sampling include patients with sepsis, organ transplants, elderly patients, patients with multiple organ failure, prolonged ICU stay, and pediatric patients. Blood sparing devices and small-volume blood collection tubes, frequent training on PBM bundles, individual settings for target hemoglobin levels, and frequency of blood sampling, as well as systematic documentation, are concepts shown to be effective, improving outcome and reducing the need for RBC transfusion. In conclusion, iatrogenic blood loss has been a well-known problem for many years. Yet not much has changed to tackle this issue. There is an urgent need to increase awareness of blood loss due to diagnostic blood sampling, and solutions such as closed-loop systems, reduction of blood sample volume, and frequency of laboratory tests should be implemented.

## Figures and Tables

**Figure 1 jcm-11-00320-f001:**
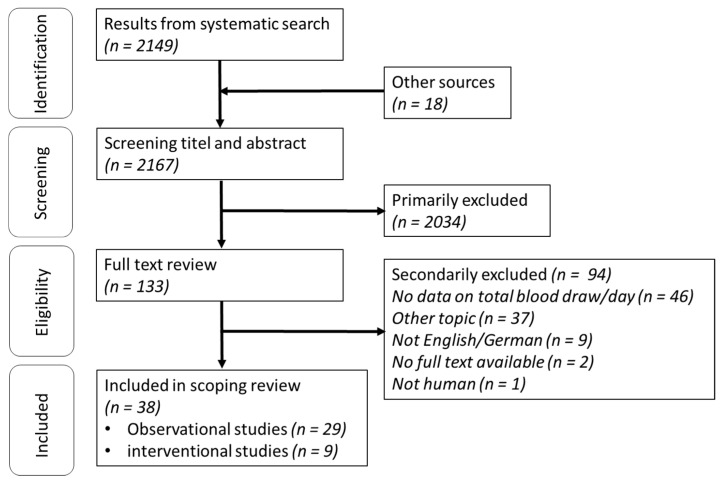
Flow chart of systematic search. *n* = number.

**Figure 2 jcm-11-00320-f002:**
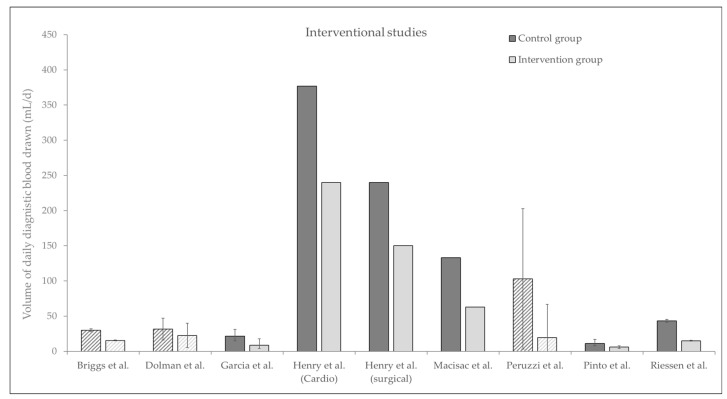
Interventional studies. Daily diagnostic blood drawn in mL/d, before and after intervention. Plain bars show median with interquartile range, and crosshatched bars show mean values with standard deviation.

**Figure 3 jcm-11-00320-f003:**
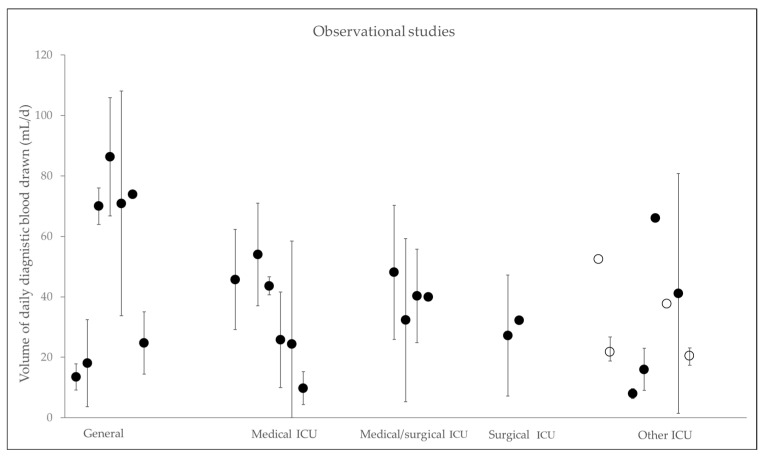
Observational studies. Daily diagnostic blood drawn in mL/d, clustered into types of ICU. Black dots show mean with standard deviation and white spots show median with interquartile range. References corresponding to Table 3: General ICU [1,9,10,46,49,50,60]; Medical ICU [27,29,47,48,51,54]; Medical/surgical ICU [11,56,57,62]; Surgical ICU [33,64]; other ICU [32,52,53,55,59,61,63,65].

**Figure 4 jcm-11-00320-f004:**
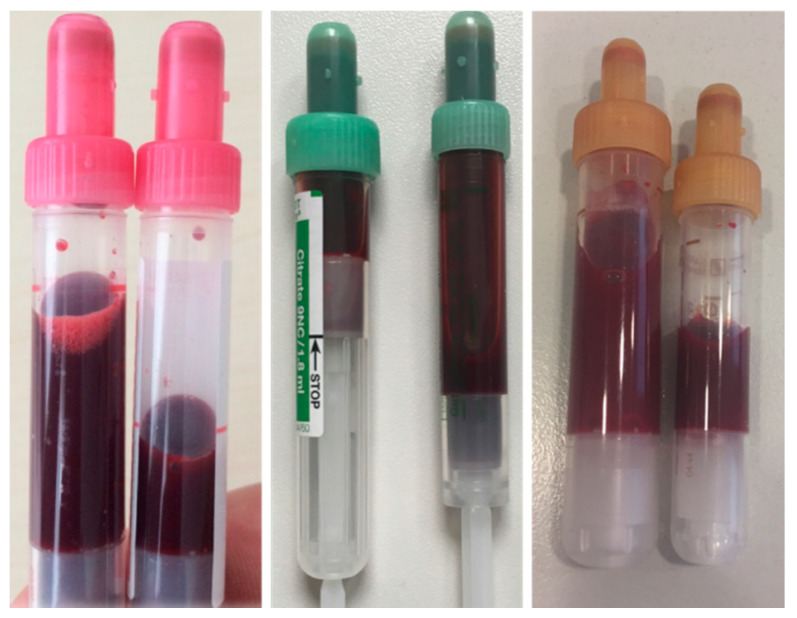
EDTA, citrate tubes, and serum chemistry with standard volume and reduced blood volume.

**Table 1 jcm-11-00320-t001:** Causes for anemia in critically ill patients.

Pathophysiological Causes for Anemia	Iatrogenic Causes for Anemia
Inflammation leads to: - Impairment of erythropoiesis - Reduction of RBC maturation and life span	Frequency and volume of phlebotomies
Endogenous kidney dysfunction with low EPO concentration	Hemolysis due to ECMO-therapy and CRRT
Altered iron metabolism	Blood volume discarded
Nutritional deficiency of iron, folate, vitamin B12	Invasive procedures
Fluid shift due to sepsis	Coagulation disorders due to pharmacotherapy
Major hemorrhages	Impaired/insufficient enteral feeding
Occult bleedings	Fluid resuscitation in septic patients
Coagulation disorders due to thrombocytopenia and liver synthesis disorders	Surgical interventions

EPO = Erythropoietin. CRRT = continuous renal replacement therapy. ECMO = extracorporeal membrane oxygenation.

**Table 2 jcm-11-00320-t002:** Interventional studies.

Author	Year	Country/Region	Study Design	Number Population (*n*)	Cohort (ICU)	Intervention	Mean Phlebotomy Volume (mL/d)	Length of Stay on ICU for Inclusion	Effects on Blood Transfusion (Transfused Patients %*)*	Days on ICU
Briggs et al. [38]	2019	AUS	C, before-and-after study	318	General	Pediatric tubes vs. control tubes	15–16 vs. 28–32	48 h	n.s. ^1,2^	/
Garcia et al. [39]	2020	USA	Ra, C	200	Medical	Pediatric tubes vs. control tubes	8.6 (4–18) vs. 21.6 (15–31)	<12 h	6% vs. 11%, n.s	2.5 (1.5–4) vs. 2.5 (1.5-5)
Macisaac et al. [40]	2003	AUS	Ra, C, unblinded	160	General	Blood conservation devices vs. control	63 (0–787) vs. 133 (7–1227) ^4^	After admission	30% vs. 17% (*p* = 0.04)	3.1 (0.2–30) vs. 2.0 (0.2–54)
Peruzzi et al. [41]	1993	USA	Pro, Ra, C	100	Medical	Blood conservation devices vs. control	19.4 ± 47.4 vs. 103.5 ± 99.9	/	26% vs. 32%, n.s	4.1 ± 3.6 vs. 4.6 ± 4.9
Pinto et al. [42]	2018	CAN	Pro, before-after design	369	Cardio	Pediatric tubes vs. control tubes	6 (4–8) vs. 11 (8–17)	/	20% vs. 28% (*p* = 0.08)	1 (1–35)
Silver et al. [43]	1993	USA	Pro, Ra, crossover comparison	31	Medical	Blood conservation devices vs. control	36.77 ^5^	<48 h	/	/
Dolman et al. [22]	2015	USA	Re, before-and-after study	248	Medical and surgical	Pediatric tubes vs. control tubes	22.5 ± 17.3 vs. 31.7 ± 15.5	48 h	4.4 ± 3.6 vs. 6.0 ± 8.2 RBC units, n.s.	/
Riessen et al. [44]	2015	DEU	Re, before-and-after study	91	Medical	Blood saving bundle vs. control	15.0 (14.3–15.7) vs. 43.3 (95% CI 41.2-45.3)	>72 h	8.0% vs. 31.7%, n.s	9.8 (8.6 to 11.3) vs. 13.2 (10.9 to 15.4)
Henry et al. [45]	1986	USA	/	20	General with cardiosurgery	Pediatric tubes vs. control tubes	Cardiology: 196 vs. 377 (234–478) ^3^ Surgical: 150 vs. 240 (147–312) ^3^	/	/	/

Median () with interquartile range. Mean ± with standard deviation. C = controlled. Re = Retrospective. Ra = Randomized. Pro = Prospective. Country codes according to ISO3166. ^1^ Short admission. ^2^ Long admission. ^3^ Mean with min and max values. ^4^ Median with min and max values. ^5^ Value not provided in publication (total laboratory blood volume in both groups of 257.4 mL/7 days).

**Table 3 jcm-11-00320-t003:** Observational studies.

Author	Year	Country/Region	Study Design	Number Population (*n*)	Cohort	Mean Phlebotomy Volume/Cumulative (mL)	Mean Phlebotomy Volume/d (mL)	Days on ICU (d)
Bedayse [46]	2010	TTO	Pro	134	General ICU	/	13.5 ± 4.3	/
Cioc et al. [9]	2015	ROU	Pro	35	General ICU	/	18.1 ± 14.4	9.7 ± 6.1
Foulke et al. [47]	1989	USA	Pro	151	Medical ICU	168 ± 18	43.6 ± 3	4.6 ± 5
Hashimot et al. [48]	1982	USA	Pro	/	Medical ICU	/	25.8 ± 15.8	/
Holland et al. [49]	2020	UK	Pro	40	General with cardiac ICU	/	86.3 ± 19.6	/
Low et al. [50]	1995	USA	Pro	25 ^1^	General ICU	/	70.9 ± 37.2 ^2^	/
Pabla et al. [32]	2009	UK	Pro	70	Acute renal medicine ward	215.8 ± 166	55.7 ± 11.23/week	23.1 ± 19.8
Thomas et al. [1]	2010	CAN	Pro	100	General ICU	/	24.7 ± 10.3	7.7 ± 6.6
Tosiri et al. [51]	2010	THA	Pro	44	Medical ICU	77.8 ± 59.2	9.8 ± 5.5	10.89
Vincent et al. [52]	2002	EU	Pro	1136	All ICU	/	41.1 ± 39.7	4.5 ± 6.7
Vinh Nguyen et al. [11]	2003	BEL	Pro	91	Medicosurgical	/	40.3 ± 15.4	7.7 ± 9.7
Witosz et al. [53]	2021	POL	Pro	36	Anesthesiology ICU	/	143.15 (121.4–161.65)/week	>7day
Andrews et al. [54]	1999	UK	Re	65	Medical ICU	/	45.74 ± 16.61	8.5 ± 8.8
Beverina et al. [55]	2021	ITA	Re	24	COVID-ICU	719 (424–1342) ^3^	21.7 (18.7–26.7) ^3^	29 (20–43) ^4^
Bodley et al. [56]	2021	CAN	Re	428 ^5^	Medical/surgical ICU	/	48.1 ± 22.2	12.2 ± 15.9
Chornenki et al. [57]	2020	CAN	Re	7273	Multicenter medical + surgical ICU	337 ± 411	32.3 ± 27.0	9.3 ± 13.4
Corwin et al. [10]	1995	LBN/USA	Re	/	General ICU	2156 ± 208 ^1^	70 ± 6 ^1^	25 ± 3 ^1^
Dale et al. [58]	1993	/	Re	14	Medical ICU	550 (50–2500) ^4,6^	/	/
Koch et al. [28]	2015	USA	Re	1921	Cardiac surgery ICU	332 (197, 619) ^7^	/	44 (24, 77) ^7^ h
Quinn et al. [33]	2019	CAN	Re	2052	Surgical ICU	145.2 ± 182.5 ^8^	27.2 ± 20.0 ^8^	5.5 ± 6.1 ^8^
Salisbury et al. [29]	2011	USA	Re, multicenter	3551	Medical ICU	173.8 ± 139.3 ^9^	24.4 ± 34.1 ^9^	/
Shaffer et al. [59]	2007	NA	Re	43	Mechanical ventilated	245 ± 213 ^10^	16 ± 7	/
Smoller et al. [60]	1986	USA	Re	31 ^11^	General ICU	798.1 ^11^	73.9 ^11^	10.8 ^11^
Tarpey et al. [61]	1990	USA	Re	26	ICU	336	66.1	5.5
Wisser et al. [62]	2003	DEU	Re	170 ^12^	Medical + surgical ICU	144 ^12^	40 ^12^	4 ^12^
Astles et al. [63]	2009	UK	/	151 ^13^	Teaching hospital ICU	/	52.4 (0–128.7) ^4^	/
Eyster et al. [27]	1973	USA	/	93	Medical ICU	/	54 ± 17	20.9 ± 7.8 ^6^
Smoller [64]	1989	USA	/	41	Surgical ICU	120.2	32.2	/
Ullman et al. [65]	2016	AUS	Cross-sectional descriptive study	50	Multiple ICUs	/	37.7 (23.1)	64.1 ± 60.9 h

Median () with interquartile range. Mean ± with standard deviation. Re = Retrospective. Pro = Prospective. Country codes according to ISO3166. ^1^ Data for subgroup patients receiving >10 RBC units. ^2^ Subgroup data: ICU patients with arterial line during the first 24 h after admission. ^3^ Mean with min and max. ^4^ Median with min and Max. ^5^ Admission days. ^6^ Entire hospital stay. ^7^ Median [25th, 75th percentiles]. ^8^ Only surgical ICU patients are presented. ^9^ Data only for subgroup with moderate to severe hospital acquired anemia. ^10^ For entire hospital stay. ^11^ Subgroup data presented: ICU patients with arterial line. ^12^ Subgroup data presented: cardiovascular surgery ICU. ^13^ Patient days.

**Table 4 jcm-11-00320-t004:** Effect of pediatric sized tubes in a 12-bed surgical ICU.

	Normal Sized (mL)	Reduced Size (mL)	Δ (mL)
EDTA	13,317	7891	5425
Citrate tubes	24,871	10,411	14,460
Serum chemistry	39,390	24,684	14,706

Results of a laboratory-chemical extrapolation of the savings achieved by reducing the filling volume of blood tubes per year with about 3700 treatment days on a 12-bed surgical ICU at University Hospital Würzburg. The filling volumes are assumed for EDTA (2.7 mL vs. 1.6 mL), citrate tubes (4.3 mL vs. 1.8 mL), and serum chemistry (7.5 mL vs. 4.7 mL). The last column shows the respective calculated amount of blood saved per year. Δ = delta (normal sized – reduced size).

## Data Availability

Not applicable.

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
