# Peer review of "Avoidable Blood Loss in Critical Care and Patient Blood Management: Scoping Review of Diagnostic Blood Loss"

_jcm, 2022, doi:10.3390/jcm11020320_

Round 1
Reviewer 1 Report
Overall, the authors present an interesting and important systematic review of existing literation on patient blood management, particularly in the ICU and Emergency department and its effect on iatrogenic anemia. No major deficiencies were noted in the manuscript as the review has been presented in a systematic manner with adequate selection of studies and appropriate inclusion and exclusion criterion. Minor suggestions are as follows:
- Avoid Abbreviations in title (e.g PBM). Please use full form in the title and when the abbreviation is first used
- In tables describing the study, rearrange the studies with prospective first, followed by retrospective, rather than random.
Author Response
We thank the handling editor and the reviewers for taking the time to review our manuscript and provide constructive criticisms and helpful suggestions.
Reviewer 1:
Overall, the authors present an interesting and important systematic review of existing literation on patient blood management, particularly in the ICU and Emergency department and its effect on iatrogenic anemia. No major deficiencies were noted in the manuscript as the review has been presented in a systematic manner with adequate selection of studies and appropriate inclusion and exclusion criterion.
Minor suggestions are as follows:
- Avoid Abbreviations in title (e.g PBM). Please use full form in the title and when the abbreviation is first used
>>>We thank the reviewer for this notification and replaced PBM with “Patient Blood Management”:
„Avoidable blood loss in critical care and Patient Blood Management“
- In tables describing the study, rearrange the studies with prospective first, followed by retrospective, rather than random.
>>>We thank the reviewer for this suggestion and rearranged the order of listed studies in the tables according to the study design starting with prospective studies, followed by retrospective studies and studies with unknown design.
Reviewer 2 Report
The authors describe studies where iatrogenic blood loss in ICU can lead to patient anaemia requiring the patient to need more RBC transfusion. They conclude that paediatric sized collection tubes may be a solution to this problem. i agree that this question should be explored more. i found the manuscript to require spelling checking and some of these i have picked up in the attached PDF.
i am wondering if other avenues other than paediatric sized tubes could also be explored such as reduced lab testing requirements?
other comments/queries are stated in the attached PDF.

Author Response
We thank the handling editor and the reviewers for taking the time to review our manuscript and provide constructive criticisms and helpful suggestions.
Reviewer 2:
The authors describe studies where iatrogenic blood loss in ICU can lead to patient anaemia requiring the patient to need more RBC transfusion. They conclude that paediatric sized collection tubes may be a solution to this problem. I agree that this question should be explored more. I found the manuscript to require spelling checking and some of these I have picked up in the attached PDF.
>>>We thank the reviewer for this advice and performed a spelling check.
I am wondering if other avenues other than paediatric sized tubes could also be explored such as reduced lab testing requirements?
>>> We thank the reviewer for this valuable thought and added the aspect and results of sharing blood samples in line 302-306.
“Educational initiatives for reducing iatrogenic anaemia should be considered, including all health care professionals and a feedback system to the ordering clinicians [77]. In addition, all laboratory values obtained should be shared with all medical disciplines to avoid redundant laboratory analyses. Furthermore, blood samples should be shared between the laboratories for reducing blood drawn.”
other comments/queries are stated in the attached PDF.
>>>We thank the reviewer for the suggestions and revised the manuscript accordingly:
- Line 19: We added the following information to the abstract:
“Therefore, this scoping review addresses the amount of blood loss during routine phlebotomies in adult (>17 years) intensive care patients and whether there are factors that need to be improved in terms of Patient Blood Management (PBM).”
- Line 38-39: We included the cut off values for severe anaemia of reference Walsh et al accordingly:
“Anaemia is a common comorbidity in intensive care patients with a prevalence of up to 98% [1], of which 40-50% suffer from severe anaemia (haemoglobin <9g/dl) [2].”
- Line 153-154: We have adopted the wording regarding excluded studies:
“A total of 94 studies were further excluded: 46 studies provided no explicit data on daily blood draws, 37 studies focused on other topics, 9 studies were excluded because of language could not be translated for the purposes of this study, for 2 studies no full text was available even after contacting the authors, and 1 study included non-human species.”
- Table 2 + 3: The abbreviation NA has been replaced by „/“ for better understanding.
- Line 177-180: We specified this in more detail and corrected the numbers.
“18 studies had a prospective design, 15 studies a retrospective design, and 5 studies had no descriptions about study design.”
- Line 184: cross reference for figure 2 is added.
- Line 194: cross reference for Figure3 is added.
Line 212-214: Literature references are added in Figure 3:
“Observational studies. Daily diagnostic blood drawn in ml/d, clustered into types of ICU. Black dots show mean with standard deviation and white spots median with interquartile range. References corresponding to Table 3: General ICU [1,9,10,48,55,56,58], Medical ICU [27,29,46, 53,54,61], Medical/surgical ICU [11,50,51,64], Surgical ICU [29,59], other ICU [32,47,49,57,60,62,63,65]- Line 211-213: The requested literature references have been added in the caption.”